# Smartcrystals for Efficient Dissolution of Poorly Water-Soluble Meloxicam

**DOI:** 10.3390/pharmaceutics14020245

**Published:** 2022-01-21

**Authors:** Rita Ambrus, Areen Alshweiat, Piroska Szabó-Révész, Csilla Bartos, Ildikó Csóka

**Affiliations:** 1Institute of Pharmaceutical Technology and Regulatory Affairs, Faculty of Pharmacy, University of Szeged, Eötvös u. 6, H-6720 Szeged, Hungary; ReveszPiroska@szte.hu (P.S.-R.); bartos.csilla@szte.hu (C.B.); csoka.ildiko@szte.hu (I.C.); 2Department of Pharmaceutics and Pharmaceutical Technology, Faculty of Pharmaceutical Sciences, The Hashemite University, Zarqa 13133, Jordan; areen.alshweiat@hu.edu.jo

**Keywords:** drying, meloxicam, smartcrystals, milling, high-pressure homogenization, nanosuspension, nanocrystals

## Abstract

Nanocrystal is widely applied to improve the dissolution of poorly water-soluble drugs. We aimed to prepare meloxicam (MLX) nanocrystals using the bead mill method, followed by high-pressure homogenization (HPH). Simple drying at room temperature (RD), vacuum-drying (VD), and freeze-drying (FD) using mannitol or trehalose as a cryoprotectant were applied to obtain dry nanocrystals. The nanocrystals were fully characterized. The MLX nanosuspension containing 5% *w/v* MLX and 1% *w/v* of Pluronic F68 showing a mean particle size (MPS) of 242 nm and a polydispersity index (PDI) of 0.36 was prepared after 40 min of premilling and 30 min of HPH. The dried nanocrystals were spherical within the nano range. DSC and XRPD confirmed the absence of MLX amorphization. The smartcrystals showed enhanced MLX release. Approximately 100% release was achieved with phosphate buffer (PB), pH 5.6, and 80% was released with PB, pH 7.4, from the freeze-dried samples. The results revealed the effects of the drying method and cryoprotectant type on the properties of dry nanocrystals. The freeze-dried samples showed the smallest particle size, in particular trehalose-based samples. On the other hand, mannitol-based dried samples showed the highest crystallinity index among all nanocrystals (77.8%), whereas trehalose showed the lowest (59.2%). These factors explained the dissolution differences among the samples.

## 1. Introduction

The poor water solubility of drugs represents the main challenge in the pharmaceutical industry [1]. The consequences of low aqueous solubility are associated with bioavailability, side effects, and delivery concerns [2,3]. Many studies have been designed to address this problem. Therefore, several techniques and approaches have been developed to enhance the solubility and dissolution rate of poorly water-soluble drugs [4,5]. Among different applied methods, particle size reduction to the nano range has proved its applicability to produce a high surface area and increase the dissolution rate of active pharmaceutical ingredients [6]. Nanocrystals can be formulated by different industrial processes [7]. In general, first-generation nanocrystal is produced by top-down or bottom-up techniques. Milling and high-pressure homogenization (HPH) are examples of top-down methods, whereas solvent–antisolvent-based precipitation and emulsion diffusion are standard examples of bottom-up methods. On the other hand, smartcrystals represent the second generation of nanocrystals. Smartcrystal technology aims to accelerate the production of nanocrystals and/or obtain smaller sizes than first-generation methods could produce.

Smartcrystals have been created by a combination of bottom-up and top-down methods. Consequently, new techniques have been introduced such as Nanoedge or H69 [8,9], H42, and H96 technologies. Combination technology (CT) has also been developed. CT combines wet bead milling and HPH [10,11]. These new technologies can maximize particle size reduction and overcome the limitations of the standard process such as long milling time and high milling speed that may lead to contamination, unwanted degradation, or amorphization of the drug that affects the stability of nanocrystals [12,13,14,15].

This process provides the advantages of employing low homogenization pressure and short processing time alongside enhanced physical stability of nanosuspensions [16]. Many poorly water-soluble drugs have been processed by CT technology for both dermal and oral applications. The smartcrystals of poorly water-soluble flavonoids such as hesperidin, rutin, and apigenin have been introduced in the pharmaceutical industry for topical administration. CT-prepared nanocrystals showed enhanced skin penetration and high stability. Rutin nanocrystals of 300 nm particle size have been produced by bead milling at 2000 rpm, followed by two cycles of HPH at 300 bar [17]. Apigenin nanocrystals have been introduced as a dermal gel. Apigenin nanosuspension of particle size of 396 nm was prepared by subjecting the bead-milled nanosuspension into HPH for one cycle at low pressure [18]. Hesperidin nanosuspensions of 599 nm particle size were produced by bead milling and five cycles of HPH at 1000 bar. Moreover, smaller than 400 nm hesperetin nanosuspensions were obtained by ultra-turrax premilling, followed by 30 cycles of HPH at 1500 bar using various stabilizers [10,19]. Apart from this, smartcrystals prepared by CT technology could be dried by freeze- or spray-drying for further processing into tablets, capsules, and fast-melts products [20].

The particle size of smartcrystals is not fully determined. Various studies report the size of smartcrystals to be lower than 100 nm. This particle size is mostly reported for intravenous injections [16]. However, a size above 100 nm is declared in dermal delivery, as mentioned previously. Different variables affect the size of smartcrystals; the method of preparation plays a significant role in it. Particles of 35.8 nm and 50 nm size were feasible using the H96 process for aprepitant and amphotericin B, respectively [21]. The reported particle size of other methods is higher than H96. Salazar et al. [22] report a glibenclamide nanocrystal of 236 nm using H42 technology. Moreover, nanocrystals in the range of 200–400 nm particle size were reported using CT. Therefore, the process and the intended use, as well as formulation parameters such as the stabilizer type and drug content, determine the particle size. The most common excipients used to produce stable nanocrystals are stabilizers such as polymers or surfactants. Size reduction is associated with high-energy surfaces that could lead to aggregation and agglomeration. Therefore, stabilizers are required for developing stable nanocrystals either by steric and/or electrostatic stabilization. The most commonly used stabilizers in nanosuspensions are Poloxamer 188, Poloxamer 407, polyvinyl alcohol, polyvinylpyrrolidone, D-α-tocopherol polyethylene glycol succinate (TPGS), cyclodextrins, and Soluplus [23,24]. Poloxamer 188, also known as Pluronic^®^ F-68, is a water-soluble synthetic triblock copolymer composed of polar chains (polyethylene oxide) and a nonpolar center core (polypropylene oxide). F68 possesses surface-active properties. F68 surrounds drug crystals, providing steric hindrance and preventing particle aggregation and growth [25]. Drying is a common practice in nanosuspensions to improve the physical and chemical stability of these systems. Spray- and freeze-drying are the most common techniques used to obtain dry nanocrystals. However, drying procedures may cause additional stress on the system, which could lead to aggregation, leading to a decrease in dissolution. Therefore, parameters of drying must be cautiously selected to ensure the redispersibility of dry powders [25,26,27].

Meloxicam (MLX), a nonsteroidal anti-inflammatory drug (NSAID), was investigated in this study. MLX belongs to class II biopharmaceutical classification systems due to its low solubility and high permeability. Moreover, MLX has limited solubility in the most common organic solvent. MLX is an oxicam derivative that forms an amphoteric molecule that exists as a zwitterion based on the used solvent. Due to low water solubility, MLX displays a slow onset of action, low bioavailability, and certain side effects [28,29]. Therefore, solubility and dissolution enhancement of MLX is the key factor to achieving adequate pharmacokinetic properties and overcoming inefficient delivery [30]. A literature review of media milling shows that as a rule of thumb, drugs of high molecular weight, low solubility, and high melting points can be processed into nanoparticles of a unimodal particle size distribution [31]. Therefore, MLX with a melting point of 254 °C and water solubility of 4.4 µg mL^−1^ showed suitability for size reduction. The low water solubility favors particle size reduction by minimizing the potential of Ostwald ripening [32].

Previous studies were performed by our team to prepare MLX as nanocrystals using different additives and milling parameters. In our earlier studies, cogrinding of MLX produced a mean particle size of 174 and 140 nm for PEG and PVP-C30, respectively [33]. Afterward, 126 nm of MLX nanocrystals was produced by combining planetary and pearl milling using polyvinyl alcohol [34,35,36]. This study aimed to investigate the feasibility of preparation of smartcrystals of poorly water-soluble MLX by a combination of bead milling and HPH as a less known CT, which could offer the advantages of applying mild conditions, including short time and low-speed milling in addition to a few cycles of HPH; therefore, the physical and chemical decomposition of MLX could be avoided. To the best of our knowledge, no previous studies have discussed the smartcrystals of MLX. Poloxamer 188 (Pluronic F68) was selected as a stabilizer. Our goal was to elucidate the effect of various drying methods on the properties of the dry powder to explore a process of best dissolution enhancement. Simple drying at room temperature, vacuum-drying, and freeze-drying with the two cryoprotectants mannitol and trehalose were applied. The size, shape, and surface morphology of the dried particles were visualized by scanning electron microscopy (SEM). The physicochemical characterization of the nanosized particles such as thermal behavior and crystalline nature were determined using differential scanning calorimetry (DSC) and X-ray diffraction (XRD). Solubility and dissolution studies were carried out.

## 2. Materials and Methods

### 2.1. Materials

Meloxicam was supplied by EGIS Ltd., (Budapest, Hungary), β-D-mannitol (M) was obtained from Hungaropharma Plc. (Budapest, Hungary), and Poloxamer 188 (Pluronic F68) was purchased from BASF (Ludwigshafen, Germany). D-Mannitol was supplied from Molar Chemicals Ltd. (Budapest, Hungary). Trehalose dihydrate was purchased from Quadrant Holdings (Cambridge, UK). Distilled and ultrapurified water was used (Milli-Q, Millipore, Merck KGaA, Darmstadt, Germany).

### 2.2. Methods

#### 2.2.1. Production of the Nanocrystal (Smartcrystal)

Second-generation nanocrystals (smartcrystals) were prepared by a combination of bead milling and HPH technologies. The particles were firstly premilled and subsequently passed through a high-pressure homogenizer [10]. In this study, 2.5 g of MLX was suspended in 1% *w/v* of F68 solution. The suspension was milled at 400 rpm rotation speed for 40 min in the milling chamber (50 mL) of a planetary mill (Retsch PM 100) (Retsch GmbH, Haan, Germany). The milling medium for this study was 50 g of 0.3 mm zirconium dioxide beads. After milling, the mixture was filtered to remove the ZrO^2^ beads, and the filtrate was diluted with 1% *w/v* F68 solution. The milled nanosuspension was further processed in HPH using a piston-gap high-pressure homogenizer (AVESTIN, Inc., Ottawa, Canada) at 1000 bar for 30 min. The final nanosuspension consisted of 5.0% of API and 1% of F68. Afterward, the nanosuspensions were dried by different methods to obtain dry nanocrystals.

#### 2.2.2. Production of Dry Nanocrystal

The nanosuspensions were dried by different methods. Drying at room temperature (RD), in a vacuum oven at 25 °C (VD), and freeze-drying with 3% *w/v* mannitol or trehalose as a cryoprotectant (FD) were used. For the freeze-drying method, the nanosuspensions were frozen to −40 °C for 24 h, followed by drying for 72 h at 0.01 mbar pressure using Scanvac, CoolSafe 100-9 Pro type apparatus (LaboGeneApS, Lynge, Denmark).

For comparison purposes, physical mixtures analogous to the prepared smartcrystals, same composition and ratio, were prepared by mixing the drug and the related excipients in a Turbula mixer (Turbula SystemSchatz; Willy A. Bachofen AG Maschinenfabrik, Basel, Switzerland) at 60 rpm for 10 min. See Table 1.

#### 2.2.3. Particle Size and Zeta Potential Measurement of the Nanosuspensions

The mean particle size (MPS), zeta potential (ZP), and polydispersity index (PDI) of nanosuspensions were measured by laser diffraction using a Malvern Nano ZS zetasizer (Malvern Instrument, UK). Water was used as a dispersant, and the refractive index was set to 1.596. The nanosuspensions were diluted by distilled water and measured at 25 °C. Twelve parallel measurements were carried out.

#### 2.2.4. Analysis of Dry Nanocrystals

##### Scanning Electron Microscopy (SEM)

The SEM images of the raw API and dry nanocrystals were captured by scanning electron microscopy (SEM) (Hitachi S4700, Hitachi Scientific Ltd., Tokyo, Japan) at 10 kV. The samples were coated with gold–palladium (90 s) with a sputter coater (Bio-Rad SC 502, VG Microtech, Uckfield, UK) using an electric potential of 2.0 kV at 10 mA for 10 min. The air pressure was 1.3–13.0 mPa.

##### X-Ray Powder Diffraction (XRPD)

The structure of the raw API and dry nanocrystals was characterized using a BRUKER D8 Advance X-ray powder diffractometer (Bruker AXS GmbH, Karlsruhe, Germany) with Cu K λI radiation (λ = 1.5406 Å) and a VÅNTEC-1 detector. The powder samples were scanned at 40 kV and 40 mA, with an angular range of 3° to 40° 2θ, at a step time of 0.1 s and a step size of 0.01°.

##### Differential Scanning Calorimetry (DSC)

The thermal analysis was performed by using Mettler Toledo DSC 821e (Mettler Inc., Schwerzenbach, Switzerland). About 3–5 mg of powder was placed into DSC sample pans, which were hermetically sealed but lid pierced. An empty pan was used as a reference in an inert atmosphere under constant argon purge. The samples were examined in the temperature interval of 25–300 °C at a heating rate of 5 °C min^–1^.

##### Saturation Solubility

The solubility of the raw APIs and dry nanocrystals was measured in phosphate-buffered solution (PBS) at a pH of 5.6 and 7.4. Excess amounts of the samples were added into 5 mL of the required PBS and stirred for 24 h at 25 °C. Then, the suspensions were filtered through 450 nm, and the concentrations were determined spectrophotometrically by UV (Unicam UV/VIS Spectrophotometer, Cambridge, UK) at λ_max_ of 363 nm.

The calibration curve of MLX was performed between 1 and 15 µg mL^–1^. The calibration curve was linear throughout the whole range tested and described by the equation A = 0.0484 Conc.+ 0.0014 (R^2^ = 0.999) at PBS of pH 7.4 and A = 0.0251 Conc.* 0.0014 (R^2^ = 0.9997) at PBS of pH 5.6.

##### In Vitro Studies

In vitro dissolution studies were carried out using a USP Type II paddle apparatus, rotating at 100 rpm at a temperature of 37 ± 0.5 °C. Drug release studies were conducted in 100 mL of PBS at a pH of 5.6 or 7.4 for 2 h. Samples that contained 0.80 mg equivalents of MLX were tested in 100 mL of dissolution media at 37 °C and 100 rpm rotation speed of the paddles. Five milliliters of the solution samples were withdrawn at predetermined time intervals, filtered (cutoff: 200 nm, Minisart SRP 25, Sartorius, Germany), and replaced by an equal volume of the fresh medium to maintain a fixed dissolution medium volume. Eventually, the amount of dissolved drug in the samples was determined spectrophotometrically by ATI-Unicam UV2-100 UV/VIS spectrophotometer at 363 nm.

## 3. Results

### 3.1. Particle Size of the Prepared Smartcrystals

MLX nanosuspensions showed a mean particle size (MPS) of 349 ± 32.5 nm and a polydispersity index (PDI) of 0.36 ± 0.001 after 40 min of milling at 400 rpm. HPH produced a further reduction in particle size. The MPS and PDI were 242.1 ± 24.2 nm and 0.36 ± 0.001, respectively. These results confirmed the impact of combination over single methods to produce smaller particle nanosuspensions [22]. The CT-produced MLX nanocrystals showed approximately 150-fold reduction compared to raw MLX of 36.54 ± 5.9 μm.

### 3.2. Zeta Potential of the Prepared Nanocrystals

ZP represents the charge of the nanoparticle. As a rule, in dispersions, the higher the charge, the more stable the nanosuspensions. The MLX-milled nanosuspension showed a ZP of –18.5 ± 4.54 mV. HPH increased the charge, as the nanosuspension showed a ZP of –30.8 ± 6.62 mV after homogenization. ZP of ±20 mV is required for the stability of the nanosuspension stabilized by steric stabilizers [37]. Therefore, MLX nanosuspensions showed evidence of stability.

### 3.3. Particle Size of the Dry Nanocrystals

MPS of the dry particles was measured after reconstitution to their original volume in water (Table 2) (Appendix A). An increase in average particle size was observed after redispersion. This increment could be attributed to the aggregation of the particles. The size varied based on the drying method and the additives (mannitol or trehalose) for freeze-drying. The freeze-drying technique produced nanocrystals of the smallest particle size. The lower MPS could be related to the effect of matrix-forming material in preventing the aggregation of the particles. However, an increase of 83 and 132 nm was observed for the trehalose and mannitol cryoprotectants, respectively. Smaller particle size was detected in the presence of trehalose compared to the size in the presence of mannitol. Mannitol tends to crystallize from the frozen solution. Thus, it provides weaker protections for the freeze-dried nanocrystals compared to trehalose that forms a glassy state. The glassy state would protect the nanocrystals from aggregation. Moreover, freeze-drying at a temperature below the glass transition temperature of trehalose could preserve the freeze-dried structure [38,39]. The PDI values of samples were lower than 0.5. Freeze-dried samples showed a more uniform distribution, as indicated by their lower PDI compared to MRD and MVD.

Stable nanosuspensions must have ZP values exceeding ±20 mV and ±30 mV for sterically and electrically stabilized nanosuspensions, respectively [7]. The reconstituted nanocrystals showed different ZP values, where MFD-T showed the highest ZP. MFD-M showed −20.9 mV, which would be enough to obtain a sterically stable nanosuspension. MRD and MVD samples displayed lower ZP values than freeze-dried samples. The higher ZP of the freeze-dried nanocrystals could be attributed to enhanced specific interaction between MLX and the polymeric stabilizers during drying [25].

In summary, the drying method and drying additives are parameters of high significance on the particle size, particle size distribution, and zeta potential of nanocrystals.

The MPS, PDI, and ZP for 1-month-stored nanocrystals at room temperature showed an increase in the particle size and a reduction in ZP (Appendix A) after redistribution in water. The increments in particle size were pronounced in MRD and MVD samples, where second peaks in the particle size plots appeared at around 10,000 nm, and the MPS values were 804.9 ± 3 and 653 ± 17.2 for MRD and MVD, respectively. On the contrary, an increase in MPS of approximately 71 and 32 nm was observed in MFD-M and MFD-T, respectively. Moreover, the ZP values for MFD-M and MFD-T were −18.6 and −22.9, respectively. The values of MPS and ZP indicated the superiority of freeze-drying with trehalose to produce stable MLX nanocrystals.

### 3.4. Morphology

The SEM images of the samples are shown in Figure 1. Raw MLX particles existed as microcrystals. However, MLX nanocrystals obtained by the CT method were spherical in the nano range. Moreover, the SEM images revealed the differences in particle size after different drying procedures. The freeze-dried nanocrystals using trehalose as a cryoprotectant were visually well separated. However, some aggregations appeared in nanocrystals dried at room temperature, as well as vacuum-dried nanocrystals.

### 3.5. Structural Analyses (DSC and XRPD)

DSC (Figure 2) was employed to investigate the thermal characteristics of the nanocrystals. The thermogram of raw MLX displayed a sharp endothermic peak at 260 °C corresponding to its melting point [40]. In PMs, the peak of MLX has been shifted toward lower temperatures. This shifting could be related to the effects of weak intermolecular bonds between the MLX and the excipients. However, such interactions have not implied incompatibility between MLX and the excipients [41]. On the other hand, the thermograms of MLX nanocrystals were similar to the corresponding PMs. The MFD-M thermogram shows a mannitol melting point at 165 °C. The trehalose containing sample MFD-T showed a peak at 195 °C, followed by a new peak at 200 °C. These peaks could be related to the anhydrous crystals of trehalose formed by freeze-drying [42]. The position of these peaks interfered with the melting point of MLX. Moreover, the presence of the amorphous form of the drug should not be excluded. Therefore, investigations by XRPD must be applied to check the crystallinity of MLX.

The X-ray diffractograms of raw MLX, PMs, and dried samples are shown in Figure 3. MLX showed distinctive peaks at 2θ values of 13°, 14.9°, 18.5°, and 25.7° [43]. The characteristic peaks of MLX were visible in the PMs besides the added patterns of the excipients. By comparing the different samples, diffractograms showed identical peak shapes for both MLX and MLX-dried nanoparticles. MFD-M showed a peak at 9.6 that suggests the presence of δ-mannitol form in all the freeze-dried samples [44,45]. On the other hand, no characteristic peaks were shown for trehalose in MFD-T, indicating the transformation of trehalose into an amorphous form during freeze-drying [46]. According to these findings, all produced nanocrystals had similar crystallinity as the raw material. Preserving the crystalline state confirms the stability of the product, as crystalline substances are physically more stable than amorphous substances.

The above finding is refuted by the calculated crystallinity indices (%X_CI_), which indicated the reduction in the crystallinity degree during nanonization (Table 3). The %X_CI_ showed that excipients affected the crystallinity degree as trehalose-based nanocrystals showed lower values than mannitol. Moreover, the drying process showed impacts on the crystallinity as MVD had lower values than MRD. Milling is a high-stress mechanical burden for the drug crystals that could lead to some amorphization [47]. This change was better evaluated by measuring the crystallinity indices rather than comparing the diffractograms. The main question is what limit should be considered to determine if the drug is crystalline or amorphous. There is a gap of knowledge in the literature regarding this issue. No data have been reported about the limits in general nor about smartcrystals in particular. Therefore, comparing the crystallinity indices with the raw drug could be beneficial. There was an obvious reduction in crystallinity of MLX in the dried nanocrystals. However, no absence of crystallinity was obtained. The reduction in crystallinity could be related to the interaction of the drug with the surfactant or cryoprotectant. Furthermore, the stress conditions of the drying method on the crystalline drug particles could contribute to the amorphization of the drug [25]. Therefore, other techniques such as DSC and Ramman spectroscopy are required for further evaluation and definition of the zones of crystallinity.

### 3.6. Solubility Studies

Table 4 lists the solubility of prepared smartcrystals and the raw drug. MLX exists in the anion form in the neutral or weakly basic solutions. Therefore, the solubility in PBS of pH 7.4 is higher than in pH 5.6 [29]. At pH 5.6, all MLX nanocrystals showed enhanced solubility compared to the raw drug. Variations appeared among the different drying methods. MLX nanocrystals dried by vacuum showed the highest solubility, followed by the room-dried nanocrystals. On the other hand, freeze-dried nanocrystals showed lower solubility than vacuum- and room-dried nanocrystals. The solubility enhancement of MRD and RVD samples could be related to the presence of pores resulting in a higher surface area [34]. These pores were detected in the SEM images of these samples (Figure 1). On the other hand, the solubility of MFD-M and MFD-T was similar.

At a pH of 7.4, the maximum solubility was achieved by freeze-dried nanocrystals with mannitol (MFD-M). The solubility improved by 2.5-fold compared to the raw MLX. MFD-T showed a slight increment in solubility due to the enhanced wettability of the MLX nanocrystals using hydrophilic trehalose. Solubility improvements of other samples were insignificant. The higher solubility of mannitol freeze-dried nanocrystals could be related to the smaller particle size shown by these nanocrystals. Moreover, this rise in solubility could be related to the affinity between MLX, F68, and mannitol to form a molecular dispersion of higher solubility at pH 7.4.

XRPD patterns of the solids after solubility analysis were similar to the nanocrystal patterns, indicating that there was no change in the form of MLX nor solvates that had occurred during the preparation of the nanocrystals (Appendix A).

As a consequence, the solubility of nanocrystals was associated with the drying method and the type of cryoprotectant.

### 3.7. Dissolution Behaviors

Based on the solubility data, the sink conditions were considered in this study. To maintain the sink conditions, saturation solubility of the drug should be at least three times the drug’s concentration in the products [48].

MLX nanocrystal products showed an enhanced dissolution rate compared to the raw drugs at PBS of pH 5.6 and 7.4. The dissolution rate depended on the drying method and the dissolution media.

The dissolution rates of MLX-related samples are presented in Figure 4. At pH 5.6, the dissolution rate was higher for the freeze-dried nanocrystals compared to the other methods. More than 90% of MLX was released after 10 min at pH 5.6 (Figure 4a). The higher release of MLX from freeze-dried samples could be related to the small particle size as MFD-M and MFD-T showed a particle size of 374.75 and 325.75, respectively. On the other hand, vacuum- and room-temperature-dried samples showed a low release. The lower MLX release from these samples could be related to their particle size and aggregation of the particles in the dissolution media. The dissolution behaviors at PBS of pH 7.4 showed an enhanced release of the nanocrystals compared to the raw drug (Figure 4b). MFD-M and MFD-T showed a rapid and high release, as 80% of MLX was released after 10 min. Apart from the freeze-dried samples, the vacuum-dried samples showed a higher dissolution rate than the room-temperature-dried samples. The dissolution studies confirmed the enhanced drug release from the dry nanocrystals due to particle size reduction, based on Noyes–Whitney equation [49]. The studies showed variations in the release according to the drying method and drying excipients. Moreover, the used excipient in the freeze-drying process could affect the drug’s wettability, hence dissolution. The dissolution rate of freeze-dried nanocrystals was different in different dissolution media. The release was almost complete at pH 5.6, but 100% release was not attained at pH 7.4. The enhanced drug release of the freeze-dried nanocrystals could be related to the amorphization of the drug during drying, small particle size, and high specific surface area, resulting in instantaneous dissolution.

## 4. Conclusions

During our work, nanocrystals of MLX were prepared using wet milling, followed by HPH. The combined technology was found applicable and effective in producing nanocrystals of improved dissolution rate of poorly water-soluble MLX of poor dissolution and amphoteric properties. CT was used successfully to prepare this agent as a nanosuspension with a particle size of 242 nm (fresh). The particle size was increased after drying depending on the drying method and the presence of drying additives. After redispersion in water, room-temperature-dried nanoparticles exhibited the largest particle size (729.3 ± 25.12 nm), followed by vacuum-dried samples (601.4 ± 15.23). Freeze-drying produced nanocrystals of the smallest particle size due to the role of the cryoprotectant in decreasing the agglomeration of the nanocrystals. Trehalose showed nanocrystals of smaller particle size than mannitol of the same concentration and under the same freeze-drying conditions. Dry smartcrystals with trehalose showed 325.75 ± 2.47 nm compared to 374.75 nm for those with mannitol. The crystalline status of products was evaluated using XRD, and the crystallinity index showed a reduction in the crystallinity of MLX. The % XCI of MLX in the nanocrystals showed values ranging between 59.2 and 77.8. The reduction in crystallinity was dependent on the drying method and the cryoprotectant in the case of freeze-drying. The smartcrystals showed distinct dissolution behaviors at different phosphate-buffered dissolution media. Almost a complete release was achieved for MLX from the freeze-dried nanocrystals at PBS of pH 5.6 and 7.4. Dissolution at pH 5.6 was more than 7.4. Moreover, the drying method played a principal role in the dissolution behavior of the nanocrystals, as room- and vacuum-drying showed a lower drug release than freeze-dried nanocrystals. The lower dissolution behaviors could be linked directly to the higher particle size resulting from these methods and the aggregation of the particles in the dissolution media.

The higher dissolution rate of prepared smartcrystals is of great interest to prepare new dosage forms of enhanced absorption and bioavailability to obtain a rapid onset of action for MLX. MLX nanosuspensions could be easily incorporated in gels for transdermal delivery, whereas dry nanocrystals could be pressed into tablets or formulated into intranasal products.

## Figures and Tables

**Figure 1 pharmaceutics-14-00245-f001:**
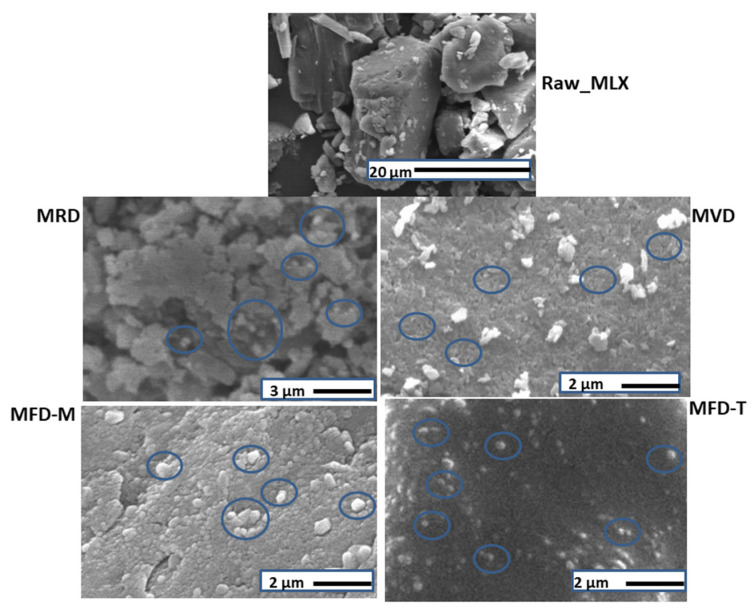
SEM images of MLX-related samples of raw MLX, MRD, MVD, MFD-M, and MFD-T.

**Figure 2 pharmaceutics-14-00245-f002:**
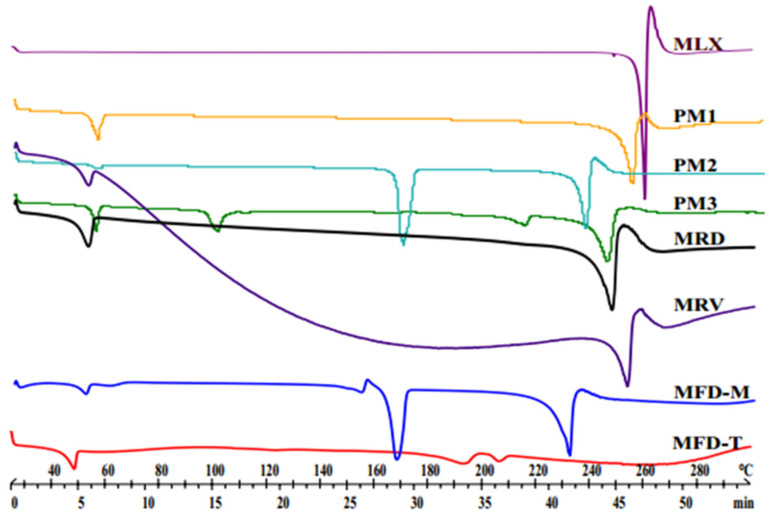
DSC thermograms of MLX-related samples of raw MLX, PM1 (MLX: F68, 5:1), PM2 (MLX: F68: M, 5:1:3), PM3 (MLX: F68: T, 5:1:3), and dried nanocrystals.

**Figure 3 pharmaceutics-14-00245-f003:**
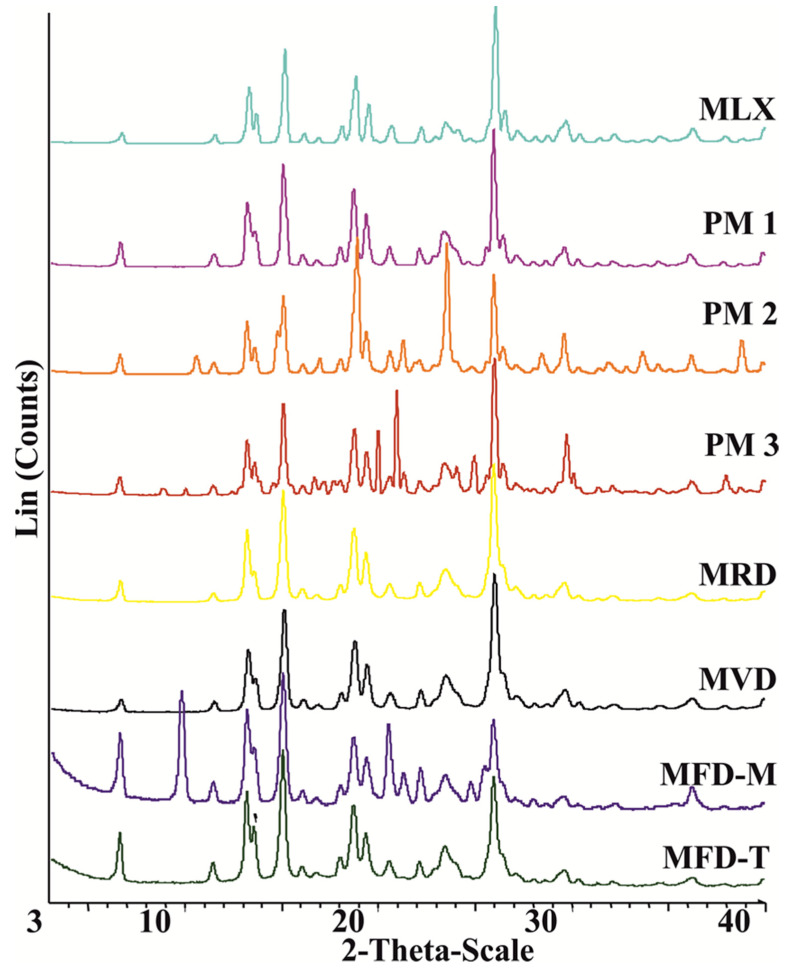
XRPD thermograms of a) MLX-related samples of raw MLX, PM1 (MLX: F68, 5:1), PM2 (MLX: F68: M, 5:1:3), PM3 (MLX: F68: T, 5:1:3), and freeze-dried nanocrystals.

**Figure 4 pharmaceutics-14-00245-f004:**
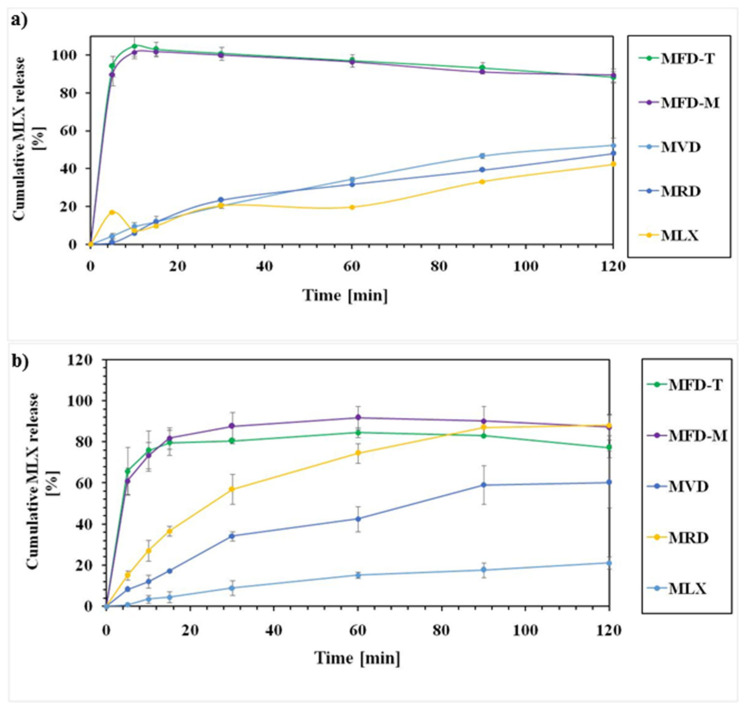
Dissolution rate of MLX-related samples at PBS of (**a**) 5.6 and (**b**) 7.4.

**Table 1 pharmaceutics-14-00245-t001:** Prepared dry nanocrystals alongside their composition and the method of drying. Notation and description of the prepared nanocrystals.

Sample Notation	Description	Content	*w:w* Ratio
MRD	MLX nanosuspension dried at room temperature	MLX: F68	5:1
MVD	MLX nanosuspension dried by vacuum oven	MLX: F68	5:1
MFD-M	MLX nanosuspension dried by freeze-drying with 3%, *w/v* mannitol (M) as a cryoprotectant	MLX: F68: M	5:1:3
MFD-T	MLX nanosuspension dried by freeze-drying with 3%, *w/v* trehalose (T) as a cryoprotectant	MLX: F68: T	5:1:3

**Table 2 pharmaceutics-14-00245-t002:** MPS, PDI, and ZP of MLX nanocrystals after redistribution in water.

Sample	MPS (nm)	PDI	ZP (mV)
**Raw_MLX**	36,540	0.77 ± 0.43	−2.3
**MRD**	729.3 ± 25.12	0.53 ± 0.12	−18.7
**MVD**	601.4 ± 15.23	0.49 ± 0.09	−18.0
**MFD-M**	374.75 ± 12.56	0.25 ± 0.02	−20.9
**MFD-T**	325.75 ± 2.47	0.26 ± 0.01	−33.1

**Table 3 pharmaceutics-14-00245-t003:** Crystallinity index of the PMs and smartcrystals of MLX.

**Sample**	% X_CI_
**MRD**	70.69
**MVD**	66.63
**MFD-M**	77.80
**MFD-T**	59.24

**Table 4 pharmaceutics-14-00245-t004:** Solubility (µg mL^−1^) of MLX and the prepared samples at PBS, pH 5.6 and 7.4 at 25 °C.

Sample	PBS	
	pH, 5.6	pH, 7.4
**MLX**	22.08 ± 1.16	477.55 ± 92.60
**MRD**	45.09 ± 1.18	492.74 ± 1.46
**MVD**	48.04 ± 6.70	478.93 ± 3.46
**MFD-M**	37.88 ± 2.81	1166.48 ± 7.57
**MFD-T**	37.88 + 2.81	588.21 ± 23.04

## Data Availability

The data presented in this study are available on request from the corresponding author.

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
