# Peer review of "Smartcrystals for Efficient Dissolution of Poorly Water-Soluble Meloxicam"

_pharmaceutics, 2022, doi:10.3390/pharmaceutics14020245_

Round 1

Reviewer 1 Report

Smart crystals for efficient dissolution of poorly-water soluble meloxicam manuscript by Ambrus et al discussed the method to alter the dissolution properties of MLX. 4 different solid forms MRD, MVD, MFD-M and MFD-T were prepared in different conditions. These solid forms were analyzed using various methods including PXRD, DSC, PSD and Zeta potential and SEM to understand their solid-state behaviours. Solubility and Dissolution properties of these 4 solid forms were studied at different pH’s (5.6 and 7.4) and these profiles were measured/analyzed by using UV spectroscopy (HPLC is suggestive however, it is absolutely fine as it is not multicomponent crystals). It is evident from the study that the lower particle-sized crystals are showing improvement in their physicochemical properties.

The authors have described all the results clearly, and it is a very simple method to improve the solubility and dissolution properties of drugs (especially drugs that belong to BCS class 2 and class 4 drugs) without altering the crystallinity of the drug.

The only thing, I would like to see in this paper is the PXRD analysis of the solids after doing solubility and/or dissolution study of the solid forms, to understand their stability after the study i.e. to check whether there is any change in their solid form (polymorphism or solvates etc). Authors can keep this information (results) in the electronic supplementary information and a few lines about this in sections 3.6 and 3.7.

Author Response

Reply to Referee comments

Reviewer 1

Thank you for your valuable comments. Below are the answers for your questions whereas modifications on the manuscript are colored by green.

The only thing, I would like to see in this paper is the PXRD analysis of the solids after doing solubility and/or dissolution study of the solid forms, to understand their stability after the study i.e. to check whether there is any change in their solid form (polymorphism or solvates etc). Authors can keep this information (results) in the electronic supplementary information and a few lines about this in sections 3.6 and 3.7.

Thank you for your remarks.

In 2005, US patent 6967248B2 claims the preparation methods for five crystal forms of MLX, which are named I, II, III, IV and V (Coppi et al., 2005). Despite its low solubility, MLX Form I is considered the most suitable one for preparing pharmaceutical products (Luger et al., 1996). According to the work of Jennifer T. and co-workers (Eur. J Pharm. Sci. 2017, 109, 374-358) they probed the occurrence of different polymorphs of MLX in raw materials. It also compared the equilibrium solubility (in raw materials), intrinsic dissolution (in raw materials) and dissolution rates (in tablets) of samples with (or without) the polymorphic contamination, which are related to bioavailability of a drug (Paulino et al., 2013, Lohani et al., 2014, Bredael et al., 2014, Resende et al., 2016). The relationship, stability and conversion involving MLX Form I and III in mixtures were also studied. They concluded that a commercially available meloxicam API that contained a mixture of polymorphs I and III. The presence of a small amount of MLX Form III (a more soluble metastable polymorph) in the API significantly affects its solubility, its intrinsic dissolution and the rate of dissolution of the meloxicam tablets.

Picture: a part of graphical abstract (Eur. J Pharm. Sci. 2017, 109, 374-358)

It could found that because of its protonable (thiazolic N) and deprotonable (hydroxyl and secondary amine) groups, MLX has pH-dependent ionization states and consequently variable aqueous solubility. Thus, MLX can adopt the cationic and anionic forms under acidic and alkaline conditions, respectively. Moreover, neutral MLX molecules can occur in either zwitterionic (enolate or amidate) or enolic forms (Luger et al., 1996, Coppi et al., 2005).

We prepared fresh samples, after it the dissolution test was carried out again. After 120 min, the media containing the dissolved and unsolved samples was dried and characterized by XRPD. We also detected the XRPD pattern for the excipients and the materials applied by the preparation of dissolution media.

Supplement Figure 1. XRPD patterns of the samples after dissolution to recrystallized the samples from the media

After the re-crystallization of the samples, compared with the result on Figure 4 in the manuscript we can detected only a very small intensity of the characteristic MLX Form III diffraction peaks e.g., at 2θ (KαCu) of 10.8; 12.7; 14.5; 16.3; 17.5; 19.4; 21.1; and 21.6° as shown in US patent 6967248B2 (Coppi et al., 2005). Form III. is a more soluble metastable polymorph, but after 24 h usually turns to Form I. It could be the reason that our final products presented Form I on XRDP patterns (Figure 4 in the text).

Figure 4. in the text

To check if there is polymorphism or solvates in the solid form, XRPD analysis of the solids after doing the dissolution study was performed. The XRPD patterns were the same as the dry samples using the different media. The figure is assigned to be in supplementary as Supplement Figure 1. The manuscript (section 3.6) was modified accordingly

The manuscript has been modified in section 3.6 “XRPD patterns of the solids after solubility analysis were similar to the nanocrystals patterns, indicating that there is no change in the form of MLX nor solvates had occurred during the preparation of the nanocrystals (Supplement Figure 1).”.

Reviewer 2 Report

Title- Smart crystals for efficient dissolution of poorly-water soluble meloxicam

In this work author had prepared meloxicam nanocrystals using bead milling followed by high pressure homogenization. In addition, effects of the drying method on the properties of the nanocrystals were evaluated. Based on the reported data author concluded that MLX nanocrystal led to enhanced release of MLX from the dosage unit. This is good work however manuscript can be improved further. Recommended to provide relevant experimental data as well as reasoning for the afforded results. My recommendation is “Major Revision” refer to my comments below.

Comments-

  1. Abstract- Author stated “The results revealed the effects of the drying method and cryoprotectant type on the properties of dry nanocrystals” suggested to briefly mention the final outcome
  2. Suggested to shorten the introduction part
  3. Production of the nanocrystal (Smartcrystals)
    • Selection of F68 stabilizer is based on literature. I assume no other polymer was tested therefore it would be good to know if author had tested concentration other than 1% of F68
    • Impact of using higher or lower concentration of F68 need to be discussed
    • For milling, speed was 400 rpm, any specific reason for choosing this speed or any DOE was performed to find critical processing parameters?
    • It is important to know the rationale behind choosing specific processing conditions, were they decided based on previous study or based on DOE which is not reported here
    • Use of zirconium dioxide beads – any comments on shredding of these beads? Because at larger scale it would be a serious problem
    • Different drug load and stabilizer should be tested to find out the processing capability to get reproducible outcome (design space) for the reported operation
  4. Particle size, XRPD, DSC, and Zeta Potential
    • Particle size of the prepared smartcrystals- if possible, suggested to provide PSD plot
    • Zeta potential- it is important to use different concentration of stabilizer to conclude if the 1% is optimal level for this formulation or not. Suggested to include such relevant data to corroborate findings
    • No stability data is provided. It is very well known that Nano formulations have tendencies to aggregate/ripening to form larger particles. Stability data (PSD, and Zeta potential) is warranted to corroborate that this approach is viable and can be successfully implemented. Suggested to include such data
    • The fact that choice of cryoprotectant can significantly influence the particle size it is highly recommended to test different concentration levels of these excipients and understand if the behavior remains same even after the accelerated stability or not?
    • Table 2 include PSD data for not dried material, if possible, include PSD plot to understand the skewness
    • SEM- suggested to provide high resolution image quality to see the changes on the particle surface. Also, to find a distinctive difference of spherical not, and particle size difference may be closer focus would help
    • DSC- suggested to include melting enthalpy value (kJ/mole) for each material. This can be helpful in determining the extent of crystallinity between starting material processed material
    • XRPD is 10% technique and amorphous <10% it will be hard to detect. Therefore, highly recommended to include DSC data with reversible heat flow in addition overlay with TGA. This will give information about the presence of Tg (if any) and TGA will help to differentiate if the Tg is associated to any solvent/water loss or its pure Tg
    • Based on Table 3 data it looks there is significant reduction in crystallinity which is essentially the amorphous (specifically for the MFD-T), so what is author’s conclusion if this CT method is good enough to form (or call) smartcrystals? Or this has to be further studied? This needs to be discussed in the manuscript
  5. Solubility-
    • At pH 5.6, is it possible that the VD and RD material has higher porosity which essentially helping to drive the solubility?
    • At pH 7.4, the solubility of VD, RD, is not significantly different than starting material. Next, MFD-T is lower than starting material. This needs possible explanation.
    • If there was no amorphous formation, then equilibrium solubility of all the material should be the same unless there is crystalline form change. If possible recommended to compare XRPD of the wet solids from the solubility study with the starting materials
  6. Dissolution- If possible suggested to do intrinsic dissolution test to understand the severity of particle (crystal) size-surface area on the dissolution (suggested to compare data from traditional dissolution intrinsic dissolution)

Author Response

Reply to Referee comments

Reviewer 2

Thank you for your valuable comments. Below are the answers for your questions whereas modifications on the manuscript are colored by red.

  1. Abstract- Author stated “The results revealed the effects of the drying method and cryoprotectant type on the properties of dry nanocrystals” suggested to briefly mention the final outcome

Thank you for your remarks. The abstract has been modified to include the effects of the drying methods. Moreover, the whole abstract was re-written to align with the journal’s regulations.

“Nanocrystal is widely applied to improve the dissolution of poorly water-soluble drugs. We aimed to prepare meloxicam (MLX) nanocrystals using the bead mill method followed by high-pressure homogenization (HPH). Simple drying at room temperature (RD), vacuum-drying (VD), and freeze-drying (FD) using mannitol or trehalose as a cryoprotectant were applied to get dry nanocrystals. The nanocrystals were fully characterized. MLX nanosuspension containing 5% w/v MLX and 1% w/v of Pluronic F68 showed 242 nm mean particle size (MPS) and 0.36 polydis-persity index (PDI) of 0.36 was prepared after 40 min premilling and 30 min HPH. The dried nanocrystals were spherical within the nano-range. DSC and XRPD confirmed the absence of MLX amorphization. The smartcrystals showed an enhanced MLX release. Approximately 100% release was achieved at phosphate buffer (PB), pH 5.6, and 80% was released at PB, pH 7.4 from the freeze-dried samples. The results revealed the effects of the drying method and cryoprotectant type on the properties of dry nanocrystals. The freeze-dried samples showed the smallest particle size. In particular, the trehalose-based ones. On the other hand, mannitol-based dried samples showed the highest crystallinity index among all nanocrystals (77.8%), whereas trehalose showed the lowest (59.2%). These factors explained the dissolution differences among the samples.”

  1. Suggested to shorten the introduction part

Thank you for your remarks. The introduction has been shortened without affecting the concept of the work. The following paragraphs have been deleted.

“In CT, low-energy milling is applied to produce nanosuspension of the drug in a surfactant solution. The milled nanosuspension is subjected to HPH for further particle size reduction. Besides, the HPH is used to lower the amount of larger crystals, improve physical stability, and produce more homogenous nanosuspensions.”

“At high concentrations (>20% w/v) F68 shows a sol-gel transition at 37°C [25–27]. However, it is used in low concentrations as a non-ionic surfactant to enhance the stability of nanocrystals [28].”

“The drug's susceptibility for size reduction could be related to its physicochemical properties, including structure (hydrophilic and hydrophobic groups), which determines inter-action with the used stabilizer. The molecular weight could also have a significant effect on the ability of drug particles to be reduced into the nano-scale. Few attempts have been made so far to understand the feasibility of nanosuspension formulation considering the drug properties. George and Ghosh [35] report that the logP of the drug has a direct correlation to the feasibility of the formation of stable nanosuspension.”

‘Various studies report that drugs of molecular weight higher than 500 g mol-1 are ideal for the preparation of nanosuspensions. MLX has a molecular weight below 500 g mol-1. However, it showed a low particle size distribution. George and Ghosh [35] report similar results as they produced the smallest particle size for the lowest molecular weight drug under investigation. Moreover,”

“The selected additives and milling procedures in these studies had a significant effect on MLX crystallinity. Amorphous and low crystallinity indices of MLX in the nanocrystals were produced using the reported methods.”

The following paragraph has been modified from “Consequently, new techniques have been introduced as Nanoedge or H69 technologies that depend on microprecipitation followed by HPH [8,9], H42 that connects spray drying and HPH, and H96 technologies used freeze-drying and HPH.” To “Smartcrystals have been created by a combination of bottom-up and top-down methods. Consequently, new techniques have been introduced such as Nanoedge or H69 [8,9], H42 and H96 technologies.”

  1. Comments related to the production of nanocrystals

  • Selection of F68 stabilizer is based on literature. I assume no other polymer was tested therefore it would be good to know if author had tested concentration other than 1% of F68
  • Impact of using higher or lower concentration of F68 need to be discussed
  • It is important to know the rationale behind choosing specific processing conditions, were they decided based on previous study or based on DOE which is not reported here. Different drug load and stabilizer should be tested to find out the processing capability to get reproducible outcome (design space) for the reported operation

Thank you for your remarks. Various stabilizers were used to prepare meloxicam nanocrystals. However, Poloxamer 188 has not been used before for wet milling meloxicam. Therefore, it was tested in this study.

Regarding concentrations, literature reported that 1% of poloxamer is the best to produce nanosuspensions regardless of the drug type ( Lestari, M.L.A.D., Müller, R.H., Möschwitzer, J.P., 2015. Systematic Screening of Different Surface Modifiers for the Production of Physically Stable Nanosuspensions. Journal of Pharmaceutical Sciences 104, 1128–1140. https://doi.org/10.1002/jps.24266). Therefore, it makes sense to use this concentration as a reference and to test other concentrations. The preliminary studies showed 1% w/v of F68 was the best concentration to produce a stable nanosuspension with a small particle size and zeta potential with this concentration was acceptable (-30 mV). The higher and lower concentrations were unable to produce nanosuspensions with acceptable particle size and zeta-potential.

Moreover, our team previously defined the critical parameters that affect the formulation of meloxicam nanosuspension. The results are found in the puplished paper “Iurian Sonia, Bogdan Cătălina, Tomuță Ioan, Szabó-Révész Piroska, Chvatal Anita, LeucuÈ›a Sorin E, Moldovan Mirela, Ambrus Rita: Development of oral lyophilisates containing meloxicam nanocrystals using QbD approach EUROPEAN JOURNAL OF PHARMACEUTICAL SCIENCES 104 pp. 356-365. , 10 p. (2017)”

In that work meloxicam nanosuspensions were prepared by high-pressure homogenization (HPH), using PVP, Poloxamer or PEG as stabilizers.

  • For milling, speed was 400 rpm, any specific reason for choosing this speed or any DOE was performed to find critical processing parameters?

The parameters of this study were empirically determined. In a previous work of our group, the parameters for wet media milling of meloxicam were determined. Milling speed less than 400 rpm was unable to produce nanosuspension (Bartos et al., 2018). Therefore, 400 rpm speed was set in our trials. However, the ability to produce nanosuspension at various time intervals was tested. The results are shown in the next table.  

The particle size of meloxicam nanosuspenion after milling time at 400 rpm

Milling time (min)

MPS (μm)

15

2.71

30

2.053

  • Use of zirconium dioxide beads – any comments on shredding of these beads? Because at larger scale it would be a serious problem
  • Use of zirconium dioxide beads – any comments on shredding of these beads?

Thank you for your remarks.

Wear could be generated during wet media milling and transferred into the related nanosuspension as contamination. The generation of wear is discussed in many studies, and the shape of zirconium wear is fully identified.

In this study, the beads were filtered, washed with alcohol, dried, and weighed. There were no significant changes in weights before and after. Moreover, the analysis of the nanocrystals showed no signs of the presence of wear. Not in the SEM images nor DSC. All the data were related to the drug or the excipients but none related to the zirconium.

  1. Particle size, XRPD, DSC, and Zeta Potential
  • Particle size of the prepared smartcrystals- if possible, suggested to provide PSD plot. Table 2 include PSD data for not dried material, if possible, include PSD plot to understand the skewness. Zeta potential- it is important to use different concentration of stabilizer to conclude if the 1% is optimal level for this formulation or not. Suggested to include such relevant data to corroborate findings

Thank you for your remarks. The zeta potential of the samples after reconstitution were measured and included in table 2. Correspondingly, the manuscript (section 3.3) was modified to consider this data “Stable nanosuspensions must have ZP values exceeding ± 20 mV and ± 30 mV for sterically and electrically stabilized nanosuspensions, respectively [7]. The reconstituted nanocrystals showed different ZP values, where MFD-T showed the highest ZP. MFD-M showed -20.9 mV, which would be enough to get a sterically stable nanosuspension. MRD and MVD samples displayed lower ZP values than freeze-dried samples. The higher ZP of the freeze-dried nanocrystals could be attributed to an enhanced specific interaction between MLX and the polymeric stabilizers during drying [7].”

In summary, the drying method and drying additives are parameters of high significance on the particle size, particle size distribution, and zeta potential of nanocrystals.

The manuscript (Section 3.3) was also modified to refer to the MPS plots in the supplementary data“(Supplement Figure 2)”

Supplement Figure 2.MPS of MLX nanocrystals after redistribution in water.

Table 2. MPS, PDI, and ZP of MLX nanocrystals after redistribution in water.

Sample

MPS (nm)

PDI

ZP (mV)

Raw_MLX

36540

0.77 ± 0.43

-2.3

MRD

729.3 ± 25.12

0.53 ± 0.12

-18.7

MVD

601.4 ± 15.23

0.49 ± 0.09

-18.0

MFD-M

374.75 ± 12.56

0.25 ± 0.02

-20.9

MFD-T

325.75 ± 2.47

0.26 ± 0.01

-33.1

  • No stability data is provided. It is very well known that Nano formulations have tendencies to aggregate/ripening to form larger particles. Stability data (PSD, and Zeta potential) is warranted to corroborate that this approach is viable and can be successfully implemented. Suggested to include such data

Thank you for your remarks on the stability of the nanocrystals. A 1-month stability was carried out (at room temprerature using excicator), then the size distribution and ZP were measured. We put it in supplementary figure. Accordingly, the manuscript (section 3.3) was modified to consider these results “The MPS, PDI, and ZP for 1-month stored nanocrystals at room temperature showed an increase in the particle size (Supplement Figure 2) and a reduction in ZP (Supplement Figure 3) after redistribution in water. The increments in particle size were pronounced in MRD and MVD samples where second peaks in the particle size plots appeared at around 10000 nm, and the MPS values were 804.9 ± 3 and 653 ± 17.2 for MRD and MVD, respectively. On the contrary, approximately 71 and 32 nm increase in MPS were observed in MFD-M and MFD-T, respectively. Moreover, the ZP values for MFD-M and MFD-T were -18.6 and -22.9, respectively. The values of MPS and ZP indicated the superiority of freeze-drying with trehalose to produce stable MLX nanocrystals.”

Supplement Figure 3. MPS, ZP and PDI of MLX nanocrystals after redistribution in water (1-month storage)

  • The fact that choice of cryoprotectant can significantly influence the particle size it is highly recommended to test different concentration levels of these excipients and understand if the behavior remains same even after the accelerated stability or not?

Thank you for your remarks. As mentioned previously, all of the parameters were selected based on our preliminary study and experimental knowledge. The stability studies showed that 3%, w/v of trehalose as a cryoprotectant produces stable nanocrystals that preserved their particle size after storage and produced enough zeta potential for physical stability.

  • SEM- suggested to provide high resolution image quality to see the changes on the particle surface. Also, to find a distinctive difference of spherical not, and particle size difference may be closer focus would help

Thanks for your suggestions. After a fresh sample preparation, new pictures were made using higher magnification as a new Figure 1.

  • DSC- suggested to include melting enthalpy value (kJ/mole) for each material. This can be helpful in determining the extent of crystallinity between starting material processed material

Thank you for your remarks. Assessing the degree of crystallinity is challenging because of the different interpretation of the crystallinity according to each technique. The Differential Scanning Calorimetry gave lower value of the degree of crystallinity compared (semi-quantitative) to the XRPD (quantitative).  Therefore, a comparative study of the crystallinity of meloxicam by using density, DSC, XRD, and Raman spectroscopy techniques will be further conducted to evaluate and define the zones of crystallinity.

The Authors referred to the need of more studies in the manuscript (section 3.5) “Therefore, other techniques such as DSC and Ramman spectroscopy are required for further evaluation and definition of the zones of crystallinity.”

     The enthalpy values were the followings:

  • XRPD is 10% technique and amorphous <10% it will be hard to detect. Therefore, highly recommended to include DSC data with reversible heat flow in addition overlay with TGA. This will give information about the presence of Tg (if any) and TGA will help to differentiate if the Tg is associated to any solvent/water loss or its pure Tg

Thanks for the suggestions. After the preparation of fresh samples, DSC and TGA measurements were carried out. We could not detected Tg and after 240 °C water loss could be seen.

  • Based on Table 3 data it looks there is significant reduction in crystallinity which is essentially the amorphous (specifically for the MFD-T), so what is author’s conclusion if this CT method is good enough to form (or call) smartcrystals? Or this has to be further studied? This needs to be discussed in the manuscript

Thank you for your remarks. In this study, a decrease in crystallinity was observed. However, full amorphization was not attained. The manuscript pointed to other factors than nanonization that could contribute to this reduction such as interaction with the used excipient. In the light of the lack of knowledge and literature about crystallinity limits, it would be speculative to determine that this method does not change the crystallinity or produce amorphous meloxicam. Therefore, more analyses are required to clarify this point.

  1. Solubility-
  • At pH 5.6, is it possible that the VD and RD material has higher porosity which essentially helping to drive the solubility?

Thank you for your remarks on the solubility of VD and RD. The SEM images of theses samples (Fig 1) show that the texture of the sample contains small pores which could increase the surface area. Section 3.6 of the manuscript has been modified to consider this explanation “The solubility enhancement of MRD and RVD samples could be related to the presence of pores resulting in a higher surface area [39]. These pores were detected in the SEM images of the samples (Figure 1). On the other hand---------“

  • At pH 7.4, the solubility of VD, RD, is not significantly different than starting material. Next, MFD-T is lower than starting material. This needs possible explanation.

Thank you for your remarks on the solubility at pH 7.4. As for MFD-T, the solubility is 585.21 ± 23.04, a deletion of the leading numbers unintentionally has occurred. The manuscript was modified to consider the correct value.

There is an increase in the solubility of MFD-T and MFD-M due to the reduction of the particle size and the presence of hydrophilic additives that increased the wettability of meloxicam. The manuscript has been modified “The solubility improved by 2.5-fold compared to the raw MLX. MFD-T showed a slight increment in solubility due to the enhanced wettability of the MLX nanocrytstals using hydrophilic trehalose.”

The reduction of particle size of meloxicam is not always accompanied with solubility enhancement. Similar results were found in a previous study where wet milling nano-sized meloxicam showed a slight increase in solubility and no order of magnitude of difference from the raw meloxicam was attained (Bartos et al., 2015).

These observations required more studies to check other factors such as polymorphism that is out of the scope of this study.

  • If there was no amorphous formation, then equilibrium solubility of all the material should be the same unless there is crystalline form change. If possible recommended to compare XRPD of the wet solids from the solubility study with the starting materials

Thank you for your remarks.

In 2005, US patent 6967248B2 claims the preparation methods for five crystal forms of MLX, which are named I, II, III, IV and V (Coppi et al., 2005). Despite its low solubility, MLX Form I is considered the most suitable one for preparing pharmaceutical products (Luger et al., 1996). According to the work of Jennifer T. and co-workers (Eur. J Pharm. Sci. 2017, 109, 374-358) they probed the occurrence of different polymorphs of MLX in raw materials. It also compared the equilibrium solubility (in raw materials), intrinsic dissolution (in raw materials) and dissolution rates (in tablets) of samples with (or without) the polymorphic contamination, which are related to bioavailability of a drug (Paulino et al., 2013, Lohani et al., 2014, Bredael et al., 2014, Resende et al., 2016). The relationship, stability and conversion involving MLX Form I and III in mixtures were also studied. They concluded that a commercially available meloxicam API that contained a mixture of polymorphs I and III. The presence of a small amount of MLX Form III (a more soluble metastable polymorph) in the API significantly affects its solubility, its intrinsic dissolution and the rate of dissolution of the meloxicam tablets.

Picture: a part of graphical abstract (Eur. J Pharm. Sci. 2017, 109, 374-358)

It could found that because of its protonable (thiazolic N) and deprotonable (hydroxyl and secondary amine) groups, MLX has pH-dependent ionization states and consequently variable aqueous solubility. Thus, MLX can adopt the cationic and anionic forms under acidic and alkaline conditions, respectively. Moreover, neutral MLX molecules can occur in either zwitterionic (enolate or amidate) or enolic forms (Luger et al., 1996, Coppi et al., 2005).

We prepared fresh samples, after it the dissolution test was carried out again. After 120 min, the media containing the dissolved and unsolved samples was dried and characterized by XRPD. We also detected the XRPD pattern for the excipients and the materials applied by the preparation of dissolution media.

Supplement Figure 1. XRPD patterns of the samples after dissolution to recrystallized the samples from the media

After the re-crystallization of the samples, compared with the result on Figure 4 in the manuscript we can detected only a very small intensity of the characteristic MLX Form III diffraction peaks e.g., at 2θ (KαCu) of 10.8; 12.7; 14.5; 16.3; 17.5; 19.4; 21.1; and 21.6° as shown in US patent 6967248B2 (Coppi et al., 2005). Form III. is a more soluble metastable polymorph, but after 24 h usually turns to Form I. It could be the reason that our final products presented Form I on XRDP patterns (Figure 4 in the text).

  • Dissolution- If possible suggested to do intrinsic dissolution test to understand the severity of particle (crystal) size-surface area on the dissolution (suggested to compare data from traditional dissolution intrinsic dissolution)

Thank you for your remarks. Unfortunately, we couldn’t find the chance to check the intrinsic solubility. However, this would be a part of our future work. 

Round 2

Reviewer 2 Report

In the revised manuscript the author has addressed all the comments and made applicable changes in the manuscript. My recommendation is “Accept”